# Diagnosis and Surgical Treatment of Drug-Resistant Epilepsy

**DOI:** 10.3390/brainsci8040049

**Published:** 2018-03-21

**Authors:** Chinekwu Anyanwu, Gholam K. Motamedi

**Affiliations:** 1Department of Neurology, Virginia Tech Carilion School of Medicine, Roanoke, VA 24016, USA; nekwusky@gmail.com; 2Department of Neurology, Georgetown University Medical Center, Washington, DC 20007, USA

**Keywords:** medically intractable epilepsy, EEG, epilepsy surgery, MRI, epilepsy, seizures

## Abstract

Despite appropriate trials of at least two antiepileptic drugs, about a third of patients with epilepsy remain drug resistant (intractable; refractory). Epilepsy surgery offers a potential cure or significant improvement to those with focal onset drug-resistant seizures. Unfortunately, epilepsy surgery is still underutilized which might be in part because of the complexity of presurgical evaluation. This process includes classifying the seizure type, lateralizing and localizing the seizure onset focus (epileptogenic zone), confirming the safety of the prospective brain surgery in terms of potential neurocognitive deficits (language and memory functions), before devising a surgical plan. Each one of the above steps requires special tests. In this paper, we have reviewed the process of presurgical evaluation in patients with drug-resistant focal onset epilepsy.

## 1. Introduction

The majority of patients with epilepsy can become seizure-free with antiepileptic drugs (AEDs) but about a third will continue to have seizures. However, patients with drug-resistant focal onset epilepsy (the most common type of epilepsy), may become seizure-free, or have significant seizure reduction, through epilepsy surgery. In particular, the efficacy and safety of anterior temporal lobectomy in patients with temporal lobe epilepsy has been established through randomized controlled trials [1].

Despite the evidence of its superiority to medical therapy in patients with drug-resistant epilepsy (DRE), epilepsy surgery is still underutilized as only a small group of these patients undergo presurgical evaluation and even a smaller group end up having surgery. It is still a common practice to try almost all available AEDs before referring patients for possible epilepsy surgery, while with proper management, diagnosis of DRE can be established within 1–2 years after seizure onset. 

Currently, out of about one million patients with DRE in the United States—the majority of whom are suffering from focal onset epilepsy and potential candidates for surgery—less than 1% are referred to epilepsy centers and only 2000 to 3000 undergo surgery. The recommended approach is to refer patients who have failed adequate trials of two AEDs, either as monotherapy or in combination, to an epilepsy center for presurgical evaluation [2,3,4]. 

This article will review proper management of patients with DRE, presurgical evaluation, and different surgical treatment options based on the current recommendations by the international league against epilepsy (IALE), and established guidelines. 

## 2. New Classification of Seizure Types

In 2017, the international league against epilepsy (ILAE) released a new classification of seizure types. This revision was based on the previous classification that has been in use since 1981. The new classification was developed to reflect the developments in our understanding of the disease since then, in particular in brain imaging, electrophysiology, and genetics. The main differences in the new classification include listing of certain new focal seizure types that used to be under the category of generalized seizures, replacing the term consciousness with awareness in description of seizures, and classifying focal seizures based on the first clinical manifestation (except for altered awareness), adding new types of generalized seizures, and adding or changing certain terms to clarity the terminology. In particular, the most common type of epilepsy, which is also potentially treatable through surgery, complex partial epilepsy, will be referred to as focal onset epilepsy with loss of awareness [5,6].

## 3. Drug-Resistant (Refractory) Epilepsy

It has been shown that 47% of patients with new-onset epilepsy can achieve complete seizure-freedom with the first AED, 13% of the remaining 53% may become seizure-free using a second agent, and only 4% with a third agent and/or polytherapy [7]. Therefore, failure of two AEDs, either as monotherapy or combination therapy, would meet the criteria for DRE [8]. However, it should be emphasized that the AEDs must have been tried adequately. An “adequate trial” includes, choosing an appropriate AED for the type of seizure being treated (e.g., failure of carbamazepine in treating idiopathic generalized epilepsy would not be considered a true failure), and titrating up the dose to the maximum tolerated dose. An AED that triggers allergic reaction or causes significant adverse effects that requires switching to another AED, may not be counted as failure either.

It is also critical to define “seizure-freedom” when assessing the success or failure of an AED. The recommended definition is sustained seizure freedom for a period 3 times the longest inter-seizure interval, or 1 year, whichever that is longer [8].

## 4. Presurgical Evaluation

The process referred to as presurgical evaluation is the most critical aspect of epilepsy surgery. All patients who are suspected of having- or properly diagnosed with- drug-resistant epilepsy should be referred to an epilepsy center for presurgical evaluation. During this process, the patient will be further evaluated to confirm the diagnosis and to classify the seizure type. 

This step is critical since close to half of patients with epilepsy have idiopathic generalized seizures which cannot be treated with resective epilepsy surgery. Only focal onset epilepsy is amenable to surgery. Therefore, only those with DRE who have focal seizures with loss of awareness (formerly, complex partial seizures), with or without secondary generalization, are surgical candidates and can complete the process of presurgical evaluation.

It is also crucial to confirm that the patient has true epilepsy and not an epilepsy mimic since patients may seem refractory for presumed seizures but may have non-epileptic spells such as syncope or psychogenic non-epileptic seizures (PNES). Therefore, the initial evaluation includes obtaining proper history, seizure semiology, and electroencephalographic (EEG) findings are role in this process. 

The ultimate goal of presurgical evaluation is to identify the epileptogenic zone (region), and to establish the safety of a prospective brain surgery, so that the resection can be performed with minimal functional impairment while achieving seizure freedom, if possible. 

Identification of seizure focus involves extensive workup including seizure characterization, lateralization, and localization through detailed evaluation of history, video-EEG (VEEG) monitoring using scalp EEG electrodes or—if indicated—via surgically implanted intracranial electrodes, different neuro-imaging modalities, neuropsychological testing, and if indicated, intracarotid Amobarbital injection procedure (IAP; Wada test).

The fundamental goal in presurgical workup includes defining the epileptogenic zone (epileptogenic region) which is the area sufficient for generation of seizures, and the minimum amount of cortex that must be resected (inactivated or completely disconnected) to produce seizure freedom. A variety of diagnostic tools should be utilized in evaluation of the various cortical zones. They include seizure semiology, VEEG recording, neuro functional testing and neuroimaging techniques with a goal to define the cortical zones involved in seizure generation and propagation. Other cortical zones of interest besides the epileptogenic zone include seizure-onset zone (cortical region that initiates clinical seizures), symptomatogenic zone (where initial ictal semiology is produced), irritative zone (where interictal spikes are generated), and functional deficit zone (cortical area that functions abnormally during interictal period) [9,10].

While the ultimate tool in detecting the epileptogenic zone is the VEEG findings (more accurately via direct intracranial recording), it can be significantly facilitated by high resolution brain magnetic resonance imaging (MRI) (structural integrity), diffusion tensor imaging (cellular integrity), magnetic resonance spectroscopy (metabolite and biochemical data), or through physiological imaging modalities such as positron emission tomography (PET) and single-photon emission computerized tomography (SPECT) scans (glucose or other ligands’ metabolism, and cerebral blood flow, respectively). The irritative zone (area that generates interictal epileptiform activities) can be detected by EEG, magnetoencephalography (MEG) and special functional MRI techniques. The seizure-onset zone (area of cortex that initiates clinical seizures) can be measured by scalp/intracranial EEG recording and ictal SPECT.

The symptomatogenic zone (area of cortex which produces the initial ictal symptoms or signs) can be determined by the seizure semiology. The functional deficit zones are measured with neuropsychological examination and functional imaging studies at the baseline (interictal period).

Despite the availability of multiple diagnostic modalities, different cases may require different diagnostic tests therefore, the evaluations process should be tailored to the individual. In particular, since a major classification in epilepsy is differentiating temporal lobe epilepsy (TLE) from extra-temporal epilepsy, it is crucial to make that determination during the presurgical evaluation. This distinction is based on the critical role of temporal lobe in neurocognitive functioning, its unique anatomy, and the clinical evidence regarding its responsiveness to surgery. 

In particular, TLE with mesial temporal sclerosis (MTS) may not require extensive diagnostic work up as it can be confirmed with concordant ipsilateral anterior-mid temporal epileptiform discharges without the need for intracranial recording. However, some patients may require intracranial (subdural) monitoring for seizure focus lateralization and/or accurate localization as well as functional cortical mapping. In general, about half of patients with TLE and almost all extra-temporal cases require intracranial EEG recording for both purposes of pinpointing the seizure onset focus and/or cortical mapping to delineate eloquent cortices for safety [11].

It should be noted that while patients who fail two appropriate trials of AEDs should be considered drug-resistant and should undergo presurgical evaluation, they may benefit from adjustments to their current AED regimens as seen appropriate. Therefore, starting the process of presurgical evaluation does not preclude using other AEDs, especially given the available variety of AEDs with different mechanisms of action and side effect profiles.

### 4.1. Scalp Video-EEG Monitoring

The standard procedure after confirming that the patient has refractory epilepsy usually starts with VEEG monitoring to further characterize, lateralize and possibly localize the seizure focus. Localization of the seizure onset zone through scalp EEG may be limited as it detects epileptiform activity that synchronizes at least 6 cm^2^ of the cortex [12]. Deeper seizure foci may not be detected via surface (scalp) electrodes. Further, scalp EEG recording is usually insufficient in extra-temporal epilepsy or even in non-lesional (normal MRI) TLE. Using extra electrodes placed based on the 10-10 international electrode system may add significant accuracy to scalp recording and in some cases avoid intracranial recording (Figure 1). 

When seizure onset zone is poorly lateralized because of alternating seizure onset lateralization, bitemporal asynchrony, or frequent bilateral epileptiform discharges, it indicates a less favorable postsurgical seizure outcome [13]. Unilateral hippocampal atrophy on MRI and concordant unilateral interictal spikes are highly predictive of concordant ictal localization [14].

Patients are often admitted for several days, depending on their seizure frequency. Most often their AEDs are tapered off in order to capture about 3–5 typical and consistent seizures. There is no consensus on tapering process but in general very rapid tapering is avoided to minimize risk of Status Epilepticus or triggering aberrant seizure onset zones. Upon the completion of scalp VEEG monitoring, some patients may not qualify for surgery; this includes patients with primary generalized seizures and those with clear multifocal seizures, although some in the latter group may qualify for intracranial recording for further elaboration. When scalp EEG cannot adequately localize the seizure focus, particularly in patients with normal brain MRI, intracranial EEG monitoring would be justified.

### 4.2. Brain Imaging Techniques

Advances in brain imaging technology have substantially improved seizure localization and surgical outcome [12]. Neuroimaging studies are not routinely indicated in all types of epilepsy. For example, patients with idiopathic (primary) generalized epilepsy syndromes such as absence epilepsy and Juvenile myoclonic epilepsy diagnosed clinically and through EEG typically do not require imaging [13]. All patients with clinical and EEG evidence of focal onset epilepsy should have brain imaging which includes brain MRI.

#### 4.2.1. MRI (Magnetic Resonance Imaging)

The principal role of MRI is to define structural abnormalities that may be the cause of seizure disorder. A high resolution brain MRI, usually referred to as Epilepsy Protocol MRI, is recommended as the modality of choice for all patients presenting with symptomatic focal epilepsies, with or without generalization. Detailed MRI sequences may be added to increase diagnostic yield depending on the etiology. Common sequences used by most epilepsy centers include thin-section (1 mm) coronal oblique T1 gradient echo, coronal oblique T2 series, high-resolution 3D sequences (sensitive to subtle cortical dysplasias or small tumors), and T2 FLAIR (fluid attenuated inversion recovery) images, performed on 3 Tesla, or higher, MRI systems [14].

Temporal lobe epilepsy is the most common type of focal epilepsy and the most common MRI finding in these patients is MTS, although this pathology may not be present at the time of seizure onset and may take many years to develop. The main radiological findings in MTS include hippocampal atrophy, internal structural derangement, and T2 hyperintensity as best seen on coronal T2 flair images (Figure 2). Other cortical epileptogenic lesions include focal cortical dysplasia (FCD), tubers, gliosis, infectious process (e.g., neurocysticercosis, abscesses), benign and malignant tumors, and inflammatory processes (Rasmussen’s encephalitis). Not all lesions are epileptogenic and patients may present with dual pathologies. In MRI-negative neocortical epilepsies, the most common pathology is FCD type 1 and 2. Type 2 FCDs are preferentially located at the bottom of the sulcus with high-resolution MRI making it possible to visually identify such in an increasing number [15,16].

#### 4.2.2. SPECT Scan (Single-Photon Emission Computerized Tomography)

Structural lesions do not always correlate with clinical semiology and EEG findings but the regional cerebral blood flow is useful tool to localize these seizures. Ictal SPECT is more sensitive and specific in temporal lobe epilepsies given that the seizures are longer. The strength of SPECT is the ability to obtain images related to regional cerebral blood flow (rCBF) at the time of seizures. 

Interictal or ictal SPECT have been used to localize the ictal onset and ictal propagation patterns and add to the evidence of abnormalities in the involved site [15,17]. Meta-analytic sensitivities of SPECT in patients with TLE have been reported as 44% (interictally), 75% (postictally) and 97% (ictally) [18] as opposed to 66% (ictally), and 40% (interictally) in extra-temporal seizures [19,20].

Ictal SPECT studies appear to correlate with outcome when there is close concordance between the area of hyperperfusion and the resected area and vice versa with surgical failure when there is poor concordance. Both ictal and interictal SPECT showing a change from hypoperfusion (interictal period) to hyperperfusion (ictal period) is more reliable than an abnormality in either stage alone. However, it has been shown that the sensitivity of ictal SPECT is higher than that of interictal SPECT because of the large CBF increase from the baseline that occurs during the ictal phase [21]. A significant limitation of SPECT scan is its logistical complexity so that a significant number of major epilepsy centers do not use it.

#### 4.2.3. FDG-PET Scan (Fluorodeoxyglucose Positron Emission Tomography)

Pet scan obtained with FDG (18 F-fluorodeoxyglucose) is considered a non-invasive technique. PET scans demonstrate hypermetabolism or increased glucose uptake during an ictal event and hypometabolism during the interictal period. When MRI is normal, PET scan may be indicated to aid in localization. It is often correlated with other studies including MRI findings, EEG neuropsychological testing and Wada test. Interictal FDG PET is known to be more sensitive than interictal SPECT in localizing extra-temporal epilepsy [22]. The changes, however, are more extensive than the structural and EEG abnormalities and may involve other regions including the ipsilateral suprasylvian and parietal regions. Interictal FDG-PET is considered to be the best imaging technique to assess the functional deficit zone (FDZ).

In TLE, interictal studies show hypometabolic areas in the epileptogenic regions in approximately 70–80% of patients (Figure 3) [23]. Ipsilateral PET hypometabolism showed a predictive value of 86% for good outcome in meta-analysis of 46 studies [24].

#### 4.2.4. MEG (Magnetoencephalography)

Magnetoencephalography maps interictal magnetic dipole sources onto MRI to produce a magnetic source image. It is a more recent imaging modality in the presurgical evaluation of focal epilepsy that has proven helpful in the detection of epileptogenic foci even in patients with inconclusive results from other non-invasive test. The additional information provided by these techniques has been demonstrated to help those patients with non-localizing MRI or extra-temporal epilepsy. MEG is in principle more sensitive to neuronal activity from superficial than deep-seated structures. It is more sensitive in detecting currents that are tangential to the surface of the scalp whereas EEG is sensitive to tangential and radial neuronal activities. MEG systems allow rapid high-resolution recordings of cortical function and dysfunction that are neither attenuated nor distorted by the skull or other variable intervening tissue layers between the scalp and brain. It primarily detects the magnetic fields induced by intracellular currents, whereas scalp EEG is sensitive to electrical fields generated by extracellular currents.

The site of surgery in 58 patients were correctly predicted by intracranial EEG in 70%, compared with MEG (52%), interictal scalp VEEG (45%) and ictal scalp VEEG (33%) [25]. MEG has shown equal effectiveness compared to EEG in terms of recording epileptic spikes with a tendency to register epileptic spikes in more patients [26].

#### 4.2.5. fMRI (Functional MRI)

While fMRI has not been established as a valid test to replace Wada memory test [27], it may be a potential complement to standard neuropsychological test to predict postoperative verbal memory outcome. It has been suggested [28,29] that it may be used to determine language dominance prior to temporal lobectomy [30,31,32,33]. Overall, besides the evidence indicating its utility as a replacement of Wada language test [34,35], to date, the replacement of the Wada memory test has proved to be more difficult [33,36,37].

The concordance between fMRI and Wada language lateralization has been shown in a number of studies reporting concordance. In one such study in 100 patients with different focal epilepsies, there was 91% concordance between the two tests. The overall rate of false categorization by fMRI was 9%, ranging from 3% in left-sided TLE to 25% in left-sided extra-temporal epilepsy [38].

#### 4.2.6. Intracarotid Amobarbital Injection Procedure (IAP, Wada Test)

A main focus of the presurgical evaluation is to estimate the risk of language- and verbal memory decline in patients undergoing anterior temporal lobectomy (ATL) on the language dominant side (usually, the left). Intracarotid Amobarbital injection procedure, commonly referred to as Wada test (after Juhn Atsushi Wada), is considered the gold standard for pre-operative lateralization of language dominance, and a measure of memory function [39]. The Wada test is also effective in predicting seizure control and degree of verbal memory decline post operatively.

Wada test is indicated in patients who may undergo temporal lobectomy. However, despite the high accuracy of the Wada test in lateralizing language and memory function, this test is associated with false negatives and false positives and needs to be performed by experienced practitioners [40]. In good hands, the test is considered reasonably safe with potential minor and major complications that have been reported to be rare, but in some reports as high as 1.09% [41]. Despite the risk, Wada test clearly remains useful in patients without clear language lateralization or with suspected atypical language lateralization, but more critically in patients at risk of postsurgical memory loss [42].

In some cases the validity of the test has been questioned [43]. Prior to the procedure, cerebral angiography is used to assess the vasculature and extent of cross-over flow to contralateral arteries. There are different anatomical variations that lead to unreliable results. EEG will be continuously recorded throughout the procedure. During the procedure, amobarbital (100–150 mg) is injected into each carotid artery to produce, one at a time, to induce a temporary disruption in function in the ipsilateral temporal lobe while the patient gets involved in a series of different language and memory related tasks. Other agents including, propofol, methohexital and etomidate have also been used [44,45,46]. 

Upon injection of the language dominant side, there will be global aphasia, while the patient would have only dysarthria after injecting the non-dominant temporal lobe. The more delicate function of Wada test is to determine the “adequacy” of the contralateral hippocampus, while shedding light on the “reserve” capacity of the ipsilateral side (seizure side). To be able to undergo anterior temporal lobectomy, ideally the patient is required to correctly identify at least 3 more objects out of a total of 8 target objects, after injecting the ipsilateral side (“split” score of 3). This would indicate enough adequacy on the contralateral hippocampus for a reasonable function ipsilateral temporal lobectomy. A suggested partial alternative to Wada test is fMRI for language lateralization however, there are no standardized methods to assess memory using fMRI. 

Currently, the test is performed in a subset of patients who appear to be at risk for clinically significant memory loss. Therefore, some patients with TLE, including those with left-sided TLE, may not need Wada test, depending on results of VEEG, MRI, fMRI, neuropsychological and PET scan results. Patients with left-sided TLE and MTS, ipsilateral hypometabolism on PET scan, low baseline performance in verbal memory and naming on neuropsych testing, and left language dominance on fMRI, may not undergo the test. However, those with nonlesional brain MRI, normal PET scan, and normal baseline verbal memory and naming on neuropsych testing may benefit from Wada test [27,47]. 

#### 4.2.7. Neuropsychological Testing

While Wada test may not be required in all patients undergoing presurgical evaluation cases, the non-invasive outpatient neuropsychological testing should be performed in all cases. It establishes a baseline neurocognitive profile that can be used for postoperative comparison, and also helps with seizure focus lateralization, localization, and postoperative outcome prediction [48,49,50]. Neuropsychological testing evaluates a variety of neurocognitive features particularly, but not limited to, in patients with TLE. These include verbal memory, visual memory, verbal fluency, and other aspects of cognitive performance. For example, a typical patient with dominant (usually left-sided) epilepsy would have significant language and memory related deficits such as verbal memory and word finding difficulties. 

On the other hand, the same type of seizure originating in the non-language dominant side would typically result in visual memory deficits. Therefore, it is crucial to both establish a baseline, and also to corroborate the results of prior work up since neuropsychological test results may further confirm—or argue against—the assumed localization and lateralization of the epileptogenic zone. The testing includes evaluation of pre-morbid intellectual function, memory function, higher executive function, language function and visuo-spartial function.

Memory and language testing contribute to the assessment of the functional integrity of the contralateral lobe. When testing suggests severe memory and language deficiencies in the contralateral hemisphere, the risk of developing postoperative memory and language impairment is high. Memory disturbances have consistently been found in patients with temporal lobe seizures because of the relationship between mesial temporal structures and memory processes [51,52,53,54,55]. 

## 5. Intracranial EEG Recording (Intracranial Electroencephalography; iEEG)

In an increasing number of patients EEG data recorded through scalp electrodes fails to clearly lateralize or localize the seizure focus. This mandates, recording the EEG directly from the brain through surgically implanted subdural surface electrodes and/or depth electrodes inserted into the brain tissue. This group of patients include those with non-lesional brain MRI, unclear seizure onset lateralization, uncertain seizure onset localization, possible bilateral or multifocal seizure disorder, suspected dual pathology, discordant noninvasive data (such as interictal and ictal EEG, structural or functional neuroimaging and neuropsychological testing or EEG findings), and seizures likely originating in eloquent cortex requiring accurate localization and cortical mapping prior to a possible resection surgery.

Implanting recording electrodes, either as depth- or grids and strips, comes with its own challenges. The location and number of hardware should be decided on an individual basis. In general, it seems safer to use minimal required hardware, the same principle that applies to AEDs. However, such general principles cannot dictate the exact surgical plan and therefore the decision-making process should be individualized in every case [56,57]. 

The use of intracranial EEG to guide surgical resections in some patients with TLE is controversial, particularly in those with presumed unilateral mesial temporal lobe foci [58]. Most patients with unilateral hippocampal atrophy with concordant scalp EEG or functional imaging studies have excellent outcomes after standard antero-medial temporal resections without intracranial monitoring with continued postoperative seizures seen in about 30% of patients [59] and some of these may be due to a dual pathology [60]. Patients with bilateral temporal lobe seizure onsets are known to have poorer surgical outcome than those with unilateral foci [61]. 

### 5.1. Subdural Electrodes

Subdural electrodes consist of small metal discs embedded in teflon or silastic sheath material and arranged in different configurations. They can be arranged in single column (strips) or in rows and columns (grid) in different sizes and configurations. These strips are implanted subdurally over the surface of brain via burr holes to insert strips, or craniotomy for grids. Large areas of the frontal, temporal, parietal, interhemispheric (para-falcine), or occipital convexities can be covered with appropriately sized grids and strip electrodes within the subdural space directly over the brain (Figure 4; Appendix A). 

Given that substantial parts of the cortex are in the depths of the sulci, subdural grids cannot effectively record activities from these areas. Stereotactic EEG (SEEG) on the other hand can assess deep sulci and gyri including mesial brain, orbito-frontal, insular and basal regions implantations without difficulty. However when mapping of the eloquent cortex is required, subdural electrodes are needed, in some occasion along with SEEG [62].

Besides their ability to record seizures directly from the cortical surface, grid electrodes are the standard method of extraoperative cortical mapping to delineate eloquent cortex. A review of 71 patients who underwent subdural grid placement found transient complications in a majority of patients with two deaths reportedly due to increased intracranial pressure [63]. The authors concluded that there was a relationship between the size of the electrode arrays and the incidence of complications. Recording delayed for more than 2 weeks may increase chances of infection and at times lead malfunction. Epidural electrodes may be used for localization of seizure onset but they are limited by the fact that they can only record from the lateral convexity of the cerebral hemispheres [64].

### 5.2. Stereoelectroencephalography (SEEG)

Recording EEG signals through surgically implanted depth electrodes provide the best coverage for deeper structures (such as hippocampus, amygdala and insula) and deep sulci. The SEEG method was originally developed by Jean Talairach and Jean Bancaud during the 1950s [65]. The first SEEG implantation was performed on 3 May 1957 at St. Anne in Paris. Its safety, along with the safety of subdural grids, has been well established [66]. The technical aspects remained essentially unchanged for several years but recent advancements has popularized the procedure. Less surgical complications have been observed with SEEG compared to subdural grids.

Risk of infection and intracranial hemorrhage have been reported in 1% and 0.8–1% of patients, respectively. In other series small hemorrhages have been reported in 5.5%, of whom only 0.9% (3/12) required surgery and no mortality were reported [67,68]. In one study, the using SEEG has been shown to confirm the epileptogenic zone in 154 patients (77.0%) with 134 patients undergoing SEEG guided resection of whom 61 patients who were followed up remained seizure free at 12 months [69,70].

The planning of SEEG implantation requires formulating precise hypotheses about the possible epileptogenic zone, seizure onset and propagation zones to be tested. It is also important to understand the functional networks involved in the primary organization of the epileptic activity while formulating a hypothesis. The trajectory passes through several target points including deep sulci, different lobes and gyri orthogonally or obliquely considering different cortical cytoarchitectural areas involved in seizure patterns. For example, in a suspected frontal lobe epilepsy case, a trajectory may pass through primary motor cortex, supplementary motor area, frontal eye field, and deep sulci.

Depth electrodes in various lengths and number of contacts are implanted using conventional stereotactic technique or by the assistance of stereotactic robotic devices through 2.5 mm diameter drill holes. 

### 5.3. Intracranial Seizure Lateralization and Localization

After implanting the intracranial electrodes, the patients are monitored in the epilepsy monitoring unit for 1–2 weeks or longer in some centers. The duration of iEEG recording is determined by seizure frequency, number of seizures needed to make a decision, and the time needed to perform mapping of the eloquent cortex.

Interpretation of the data is based on EEG pattern recognition as well as clinical semiology (symptomatology). An accurate interpretation of intracranial ictal EEG requires appropriate electrode coverage of potential epileptogenic areas. A key point is to have placed the iEEG electrodes in proper locations i.e., at—or in close vicinity to—the epileptogenic zone and areas around it. Besides the more common ictal EEG patterns such as buildup of rhythmic repetitive sharp or spike discharges (Figure 5), there is increasing evidence that high frequency oscillations (HFOs) that may be detected visually at seizure onset, represent the epileptogenic zone. Interpretation may be difficult in some cases since frequency range differs from one cerebral area to another because of neurophysiological properties of the structure and from one etiology to another [71]. 

Bursts of HFOs may also be recorded during interictal periods and may be a marker of epileptogenicity if the same bursts will develop into sustained fast activity at seizure onset. Sometimes ictal baseline direct current (DC) shifts may precede the fast ictal discharges [72].

A specific interictal pattern in FCD is the occurrence of fast discharges combined with almost continuous interictal spiking in the same region [73]. In localization of seizure onset, the most reliable part is the first clear synchronizing electrical change seen in a limited number of electrodes [74] that precedes the onset of clinical semiology. Mesial temporal lobe seizures may also begin as low-voltage, high-frequency discharge without preictal spiking [75].

### 5.4. Electrical Stimulation (Cortical) Mapping

The purpose of cortical mapping is to delineate the borders of eloquent cortex. This procedure is performed in patients whose seizure onset zone is located in the vicinity of language or motor cortices. The purpose is to avoid removing such cortical areas hence functional deficits. An alternating polarity square wave pulse pair stimulation (0.3 msec in duration, at 50–100 Hz, lasting 3–5 s) is applied to a pair of electrodes located in areas near the border of presumed eloquent cortex (Figure 6). Caution must be taken when stimulating cortex as a seizure may be provoked particularly when stimulating highly epileptogenic cortical regions such as primary motor cortex and mesial temporal structure. A provoked seizure is usually heralded by after discharges (Figure 7). [76].

## 6. Epilepsy Surgery

Following the completion of presurgical evaluation and localization of the epileptogenic zone as well as establishing the safety of a resection surgery in the individual patient, the proper surgical option can be decided. 

Depending on the seizure type, seizure localization, presence of absence of an identifiable pathology on brain imaging (lesional vs. nonlesional cases, respectively), and patient’s functional baseline, different surgical approaches and methods can be used. These include potentially curative surgeries i.e., ATL (using different variations in technique), and resection of the epileptogenic zone (often extra-temporal), or hemispherectomy and palliative surgical approaches such as corpus callosotomy.

Given the higher prevalence of temporal lobe epilepsy and its distinct features i.e., its critical role in basic cognitive functioning and its unique anatomy, epilepsy surgery is commonly categorized and discussed as two separate categories of temporal- vs. extra-temporal. 

Temporal lobe epilepsy surgery most often is performed as anterior temporal lobectomy and includes the removal of anterior and mesial structures which naturally raises concerns about postoperative language and memory deficits [1]. Therefore, as discussed earlier, specific diagnostic tests have to be performed to address those concerns before a safe surgical method can be devised. 

Extratemporal epilepsy surgery on the other hand, involves the removal of an identified seizure focus as long it does not overlap with eloquent cortex. Generally, well delineated lesions such as cavernous angioma, and FCD are more amenable to resection surgery with better outcome in lesional cases of extratemporal epilepsy than nonlesional cases [77]. 

In general, a complete resection of the epileptogenic brain region provides higher chances of seizure freedom but the risks of postoperative deficits would increase with the extension of resection. Therefore, the extent of resection should be weighed against such risks and individualized in every case. 

While traditionally the mainstay of surgical treatment has been open brain resection, recent innovations have allowed for less invasive ablative techniques such as MRI-guided laser interstitial thermotherapy (LITT). A brief review of the different technical approaches and methods follows. 

### 6.1. Surgery of Temporal Lobe Epilepsy

Anterior temporal lobectomy (ATL) is the most commonly performed surgery for the treatment of refractory temporal lobe epilepsy. Most patients who undergo ATL have concordant findings with MRI, VEEG, PET scan, and Neuropsychological testing. This procedure is performed using a variety of methods. The standard (en bloc) resection includes 3–6 cm of anterior temporal neocortex and 1–3 cm of mesial structures (amygdala and hippocampus) in the resection, with more limited resections performed on the language dominant side. 

Selective amygdalo-hippocampectomy (SAH), is another modified surgical method that was developed to avoid the resection of lateral (neocortical) temporal tissue to minimize language deficits. The SAH is performed by accessing temporal horn through an incision in the middle temporal gyrus and selectively removing mesial temporal structures leaving the neocortical region intact [78]. Approximately 70% of properly selected surgical candidates become seizure-free following ATL, and the majority of remaining group benefit significantly by achieving seizure reduction and improved quality of life (QOL) [79,80].

### 6.2. Laser Ablation Surgery

Another method of operation used in temporal lobe as well as extra-temporal epilepsy is a less invasive procedure that can ablate epileptogenic focus that avoids craniotomy. Laser ablation surgery has been effectively applied in lesional and non-lesional cases including MTS, FCD, failed prior open surgery, and on deeper lesions that are inaccessible to open surgery.

This procedure has the advantage of selectively targeting small lesions responsible for seizures, with less pain and shorter hospitalization. The goal of an MRIgLITT (MRI-guided laser interstitial thermal therapy) system is to necrotize soft tissues through interstitial irradiation or thermal therapy under MRI guidance. Various pathologies have been treated with this technique including MTS, hypothalamic hamartoma, cavernous angioma, malformations of cortical development, and tuberous sclerotic lesions [81,82].

Mesial temporal epilepsy is also suitable for MRIgLITT. A history of prior laser ablative surgery would not preclude further open resection or repeat ablation procedure. Its safety and accuracy, in patients with difficult-to-localize seizures has been shown [83], and its efficacy has been reported as comparable to that of open surgery but with less morbidity [84,85] including in older patients [86]. Evidence suggests that selective surgical approaches for mesial temporal lobe epilepsy lower cognitive risks compared to the standard ATL. [87,88].

### 6.3. Corpus Callosotomy

Corpus collosotomy is a palliative surgical procedure devised to alleviate debilitating seizures in individuals with no focally resectable lesion [89]. Disconnection of the corpus callosum decreases the severity and frequency of rapidly propagating secondarily generalized focal seizures in patients who are not otherwise good surgical candidates particularly those with drop attacks (tonic or atonic seizures).

### 6.4. Multiple Subpial Transection Procedure

This technique introduced by Morrell in 1989 is primarily used for treatment of refractory epilepsies where resection is risky such as cases of seizure foci in the vicinity of, or with overlapping, eloquent cortex. The technique leaves the vertical columnar arrangement of the cortex intact thereby preserving function while preventing horizontal propagation of the seizure discharge. The procedure has been shown to be of value in cases of epileptogenic foci involving an eloquent cortex but less effective than excision surgery [90].

### 6.5. Hemispherectomy

Hemispherectomy (FH) is used mainly in children with refractory focal epilepsies confined to one hemisphere but also rarely in adults. Two main surgical techniques include anatomic hemispherectomy (which aims to resect all cortical tissue on one hemisphere along with various amounts of subcortical tissue), and functional hemispherectomy (which consist of more disconnection and less resection than the anatomic hemispherectomy. 

Hemispherectomy is most commonly used in patients with perinatal strokes, Sturge-Weber syndrome, Rasmussen encephalitis, Infantile Hemiplegia Seizure Syndrome (IHSS), and hemimegalencephaly. Surgical candidates are selected based on a number of factors including severity of the underlying epilepsy, neurological condition, natural history of the underlying disease, and patient’s age. Serious late post-op complications including superficial cerebral hemosiderosis and hydrocephalus can occur although modifications of the procedure have decreased these complications [91,92].

## 7. Non-Resective Surgical Treatments

Patients who are determined to be unqualified for seizure focus removal via resective surgery, such as those with multifocal seizures and those with overlapping seizure onset zone and eloquent cortex, may benefit from other surgical options. Attempts to control seizures through electrical stimulation started in 1960s have resulted in a few available treatment modalities starting with vagus nerve stimulation (VNS) in 1980s. Currently, there are two such procedures approved in the United States and one other approved elsewhere. These surgically implanted modalities do not require removal of brain tissue. 

### 7.1. Vagus Nerve Stimulation (VNS)

This is an adjunctive therapy approved by the Food and Drug Administration (FDA) for refractory focal onset epilepsies with or without secondary generalization in patients 4 years or older. The VNS system is comprised of a battery generator programmed to deliver intermittent electrical stimuli via the left vagus nerve to the brain. The generator is implanted in the left upper chest and is tunneled under the skin to the vagus nerve. A retrospective review of efficacy of VNS in 48 patients with intractable partial epilepsy showed progressive decrease in mean seizure frequency by 26% after 1 year, 30% after 5 years, and 52% after 12 years [93].

### 7.2. Responsive Neurostimulation (RNS) 

The RNS system was approved by the FDA in 2013 for medically refractory focal epilepsy. It provides cortical stimulation in response to ictal discharges detected by the RNS device. This is a programmable neurostimulator which is cranially implanted and connected to one or two depth and/or subdural cortical strip electrodes placed over the seizure foci. The Neurostimulator continually senses electrocorticographic activity and when detects onset of an ictal discharges, provides brief pulses of stimulation in response (closed-loop system).

The pivotal randomized study of 191 subjects showed a progressive reduction in the frequency of total disabling seizures in the treatment group (41.5%) compared to the sham group (9.4%) in the final months of the 2 year study [94].

### 7.3. Stimulation of the Anterior Nucleus of Thalamus (Deep Brain Stimulation, DBS)

In a multicenter trial, that evaluated stimulation of bilateral anterior nuclei of the thalamus, the treatment group had 29% greater reduction in seizures compared to the control group in the first month, and at least 50% seizure reduction in 54% of patients on 2 years follow up [95]. Stimulation of the anterior nucleus of thalamus has been approved as adjunctive treatment for drug-resistant focal epilepsy in adults, in the European Union since 2010, but its approval in the by FDA for use in the United States is still pending.

## 8. Postsurgical Outcome

Postsurgical outcome in epilepsy surgery is mainly assessed based on seizure-freedom or seizure reduction, neurocognitive functioning, and change in overall QOL. More detailed elements in measuring outcome include factors such as work eligibility, driving, etc. Of particular concern, are language and verbal memory function following temporal lobe surgeries. 

Interestingly, despite the evidence supporting the efficacy and safety of epilepsy surgery, until 2001 there was no such evidence based on randomized, controlled trials (RCT). The first RCT included 80 patients with TLE randomly assigned to surgery (ATL), or continuing medical therapy, for one year to compare both seizure control outcome, and quality of life (QOL), disability, and mortality.

The study showed, for the first time, that after one year follow up, 58% of patients in the ATL group were seizure- free compared to only 8% in the medical therapy group (*p* < 0.001). Further, those in the surgical group had less frequent focal seizures and significantly better quality of life compared to the non-surgical group (*p* < 0.001 for both comparisons). In the surgical group 4 patients (10%) had mild language and memory deficits against one dead in the non-surgical group [1].

Since postsurgical outcome significantly depends on seizure type and seizure onset location, different types of epilepsies and surgical approaches are to be studied separately. A summary of a few selective outcome studies, including particular surgical methods used, follows.

### 8.1. Mesial Temporal Lobe Epilepsy with Hippocampal Sclerosis

#### 8.1.1. Standard (en Block) Anterior Temporal Lobectomy (ATL)

Patients with MTLE and MTS can achieve long-term seizure control following standard ATL however the risk of seizure recurrence after ≥2 years exists. A review of 50 consecutive cases of ATL in patients with MTLE and MTS reported 82% seizure-freedom rate at 12 months, 76% at 24 months, and 64% at 63 months (mean follow up 5.8 years). The improvement in seizure control was associated with improved long-term QOL. The study also indicated reduction in AED as the major risk factor for seizure recurrence [96]. 

A more recent study of a similar group of patients undergoing standard ATL and followed for mean 6.7 years found 89% seizure-freedom, and significant improvement—defined as Engel Class I or II in 94%. Complete seizure freedom was seen in 103 patients (89%), and Engel Class I or II outcome (free of disabling seizures, or rare disabling seizures, respectively) in 94%. This study also determined concordance rates between the final resection site and the following factors: VEEG (100%), PET (100%), MRI (99.0%), and Wada test (90.4%). The lowest concordance was found with SPECT (84.6%), and neuropsychological testing (82.5%). Interestingly, a strong Wada memory lateralization was associated with excellent long-term seizure control, and vice versa, i.e., low disparity in scores between the sites predicted continued seizures [97].

#### 8.1.2. Selective Amygdalohippocampectomy (SAH)

Patients with MTLE and MTS who undergo this technique may also achieve similar outcomes comparable to ATL with more extensive (neocortical temporal) resections [98]. While SAH was originally introduced as a method that could minimize neurocognitive deficits by preserving the neocortical temporal cortex, the controversy about its outcome involves neuropsychological deficits rather than seizure-freedom rates, as compared to standard ATL. While there is some evidence of SAH resulting in better cognitive function than ATL [99]. SAH performed in patients without adequate memory reserve can result in verbal memory deficits in dominant temporal lobe cases [100]. This topic has been reviewed by Schramm who concluded that there was considerable evidence of improved neuropsychological outcomes in SAH [101]. 

## 9. Neurocognitive Deficits Following ATL

At the presurgical baseline, patients with epilepsy, in particular those with TLE, often have some degree of cognitive impairment which may be further affected by surgery. It has been shown that larger temporal lobe resections provide better seizure control outcome however, they are more likely to induce cognitive deficits since larger resection risks including more of the functional tissue [102]. 

To further quantify association between the extent of temporal lobe resection and cognitive outcome a group of 47 right-handed patients with left MTLE who underwent ATL was studied. Cognitive changes were compared between limited resection (1–2 cm for mesial, and ≤4 cm for neocortical), and more extensive resection (>2 cm for mesial, and ≥4 cm for neocortical). There were no differences in cognitive outcome between the groups however, a negative correlation was found with age at seizure onset [103]. 

## 10. Conclusions

Drug-resistant epilepsy, defined as failure of two anti-seizure medications to completely control seizures, can be effectively treated with surgery. Therefore, proper use of diagnostic methods to determine a patient’s eligibility for surgical treatment i.e., presurgical evaluation, is of utmost importance in managing patients with DRE. This process, as discussed in this article, also determines what type of surgical approach would be the safest and the most beneficial approach to the particular patient. With further developments in diagnostic and therapeutic methods, both presurgical evaluation and the surgical methods used, will improve constantly.

## Figures and Tables

**Figure 1 brainsci-08-00049-f001:**
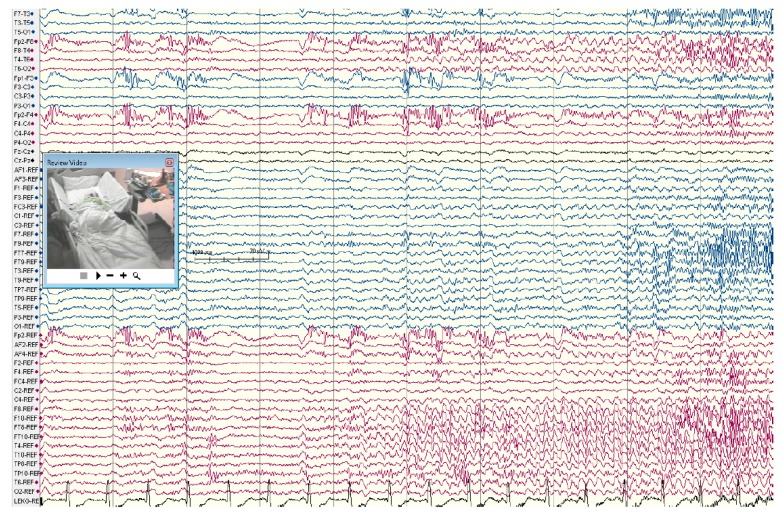
Right temporal lobe seizure onset in a patient with right hippocampal sclerosis and temporal lobe epilepsy (TLE). The EEG shows buildup of rhythmic repetitive sharp theta discharges arising in the right anterior-mid temporal regions. Scalp recording during long-term monitoring using extra electrodes placed according to the 10-10 electrode system for clear distinction between frontal and basal-lateral temporal head regions.

**Figure 2 brainsci-08-00049-f002:**
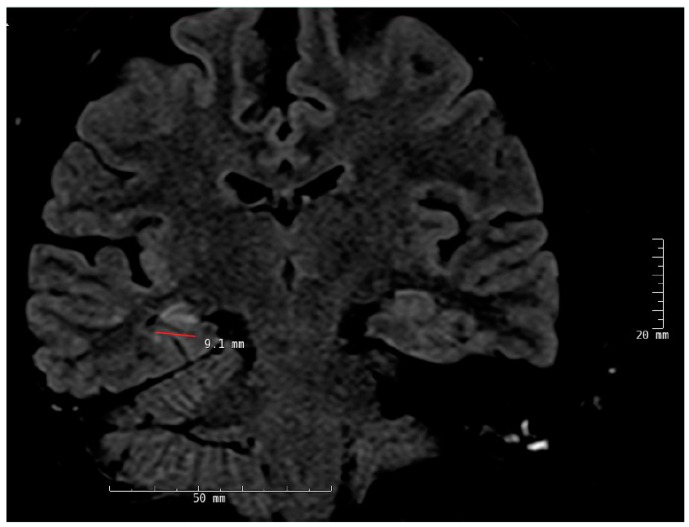
Right mesial temporal sclerosis (MTS). There is atrophy and increased signal intensity in the CA1-3 regions of the right hippocampus. Coronal view; fluid-attenuated inversion recovery (FLAIR) sequence.

**Figure 3 brainsci-08-00049-f003:**
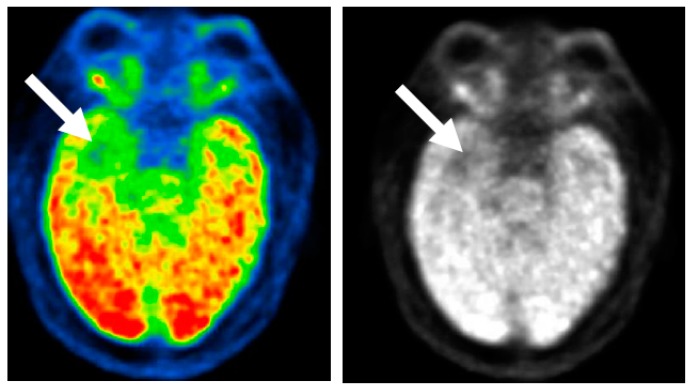
Interictal FDG PET (fluorodeoxyglucose positron emission tomography) scan. Decreased glucose uptake in the right temporal area (arrows) compared to the left side, in a patient with right mesial temporal lobe epilepsy (MTLE).

**Figure 4 brainsci-08-00049-f004:**
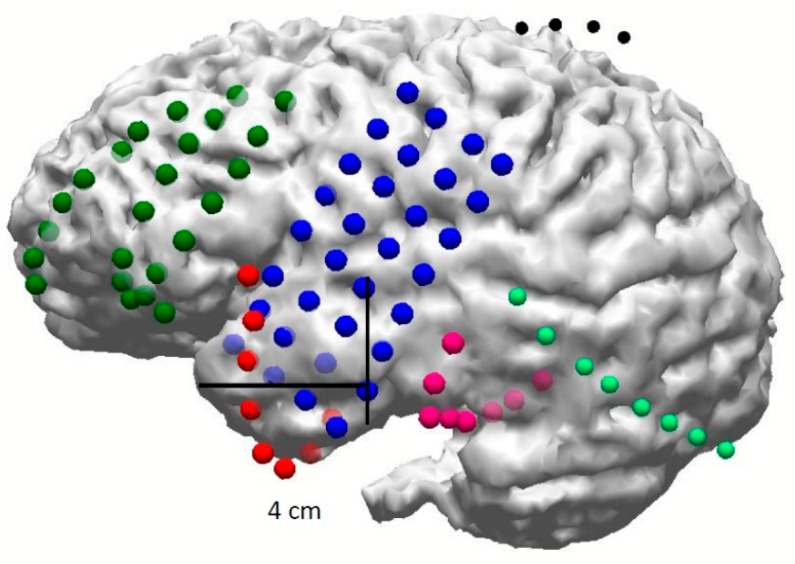
3D rendering of baseline brain magnetic resonance imaging (MRI) and surgically implanted subdural electrodes (grids and strips).

**Figure 5 brainsci-08-00049-f005:**
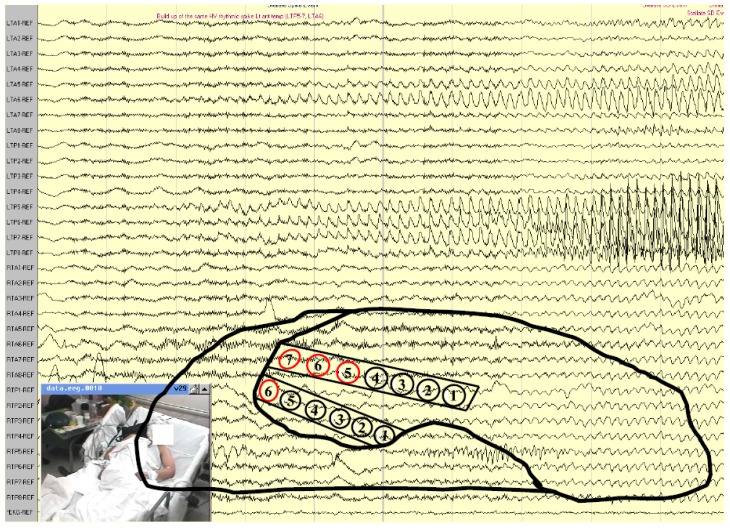
Ictal onset EEG recording through two 1 × 8 subdural strips placed over the basal temporal region in a patient with left temporal lobe epilepsy (TLE). Buildup of rhythmic sharp discharges initiated at a few electrodes (red circles) before rapid propagation to other electrodes and generalization.

**Figure 6 brainsci-08-00049-f006:**
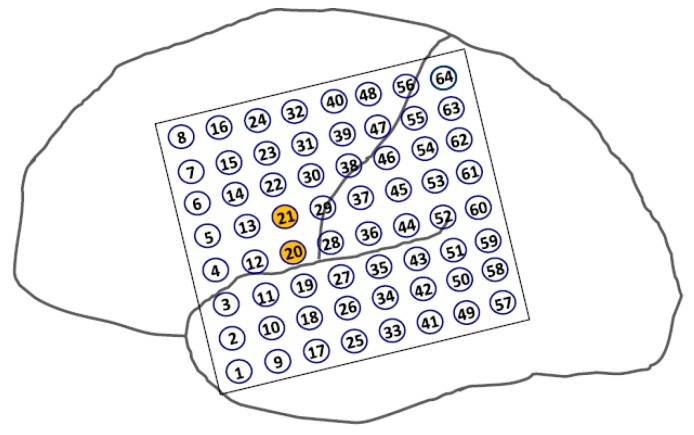
Stimulation cortical mapping. Electrodes 20 and 21, located in the prerolandic motor cortex, are stimulated based on a particular paradigm including slow titration of current density. The patient will be observed for any motor activity in the corresponding body part (face and hand) while engaged in a conversation or reading out loud.

**Figure 7 brainsci-08-00049-f007:**
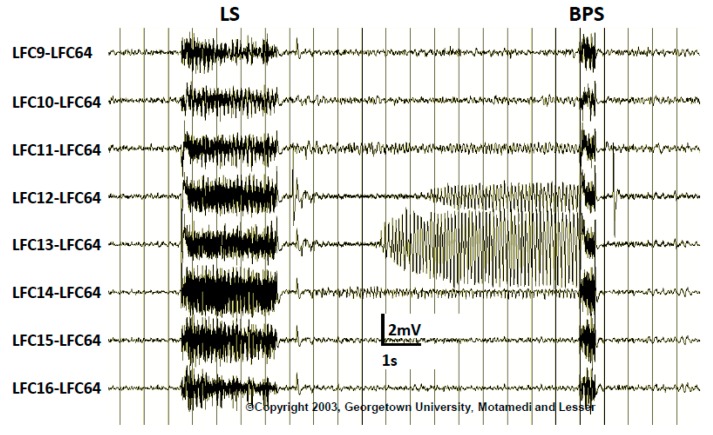
During stimulation mapping the pulse stimulus (LS, localization stimulus) may induce afterdischarges (AD). In some cases, a brief pulse stimulus (BPS) may successfully terminate the AD and prevent a clinical seizure.

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
