# Peer review of "Diagnosis and Surgical Treatment of Drug-Resistant Epilepsy"

_brainsci, 2018, doi:10.3390/brainsci8040049_

Round 1
Reviewer 1 Report
Dear Chinekwu and Gholam:
I enjoyed reading your comprehensive review of Diagnosis and Surgical Treatment of Epilepsy manuscript. Here are a few recommendations you should consider:
1) As the field of epilepsy surgery advances, minimally invasive theraputic and diagnostic procedures are being used more often. Therefore, I suggest softening the language used for epilepsy surgery by switching the word invasive recording to intracranial recording. In our center, the majority of our case are SEEG, which is minimally invasive.
2) I like the fact that you are bringing the new classification to your work. I suggest switching the parenthesis in line 63 to: refractory focal seizures with loss of awareness (complex partial seizures in previous classification).
3) Need a reference to this fact: Half temporal lobe epilepsy and all extra temporal lobe epilepsy will need invasive EEG recording. You can also say Non-lesional MRI (more specific).
4) Suggest deleting reference 84:
The ROSE trial was discussed at the AES in Huston in 2016. The 3 year ROSE trial was unable to establish the non-inferiority of radiosurgery – indeed, it actually appeared to be inferior to surgery. More patients were seizure-free during the last year of the trial if they had surgery, rather than radiosurgery (78% versus 52%). But neither conclusion could be drawn given the study's lack of statistical power. In fact, recruitment had to be stopped.
"It appears that stereotactic radiosurgery is inferior, but the low power of the study makes it impossible to verify that conclusion," lead investigator Nicholas Barbaro, MD, of Indiana University, said during a hot topics session. "Many people will take that conclusion from this, but I caution you not to reach too far from very limited data."
Mark Quigg, MD, of the University of Virginia, who presented the main findings, noted that the power to demonstrate non-inferiority was 41%, "meaning the probability of us getting the wrong answer would be 59%."We were hamstrung by the low power of the study," Quigg said.
5) In our experience MST is not as effective as excision surgery. Collective Data Supports Efficacy of Multiple Subpial Transection: Epilepsy Curr. 2002 Jul; 2(4): 108., doi: 10.1046/j.1535-7597.2002.t01-1-00039.x PMCID: PMC321030
6) FDA Approves VNS Therapy Treatment for Drug-Resistant Epilepsy in Children as Young as Four Years Old Expanded access to VNS Therapy treatment offers new hope for children with drug-resistant epilepsy, London, June 29, 2017. Need to adjust your statement to include adult and pediatric population.
More suggested detailed editing (including medical and language editing) for your consideration.
Line | Existing | Suggestion | Reason |
63 | Complex partial seizures (focal seizures with loss of awareness). | refractory focal seizures with loss of awareness (complex partial seizures in previous classification).
| To emphasize the new seizure classification and for better education of the new generation |
69 | invasive subdural recording | intracranial monitoring | More broad including SEEG |
70 | Add non-lesional before TLE | ||
70 | Add functional before cortical mapping | ||
71 | Invasive | Intracranial | Make it more surgically appealing if SEEG is being used which is minimally invasive |
76 | (where seizure semiology is triggered) | (where ictal semiology is produced) | |
77 | Readily propagated to | Are generated | |
77 | Add (functionally abnormal cortex) | ||
79 | Delete denied as | ||
85 | Functional MRI(fMRI) | Special functional MRI technique | Need detecting interictal discharges |
86 | Invasive | Intracranial | |
92 | Video-EEG | Scalp video EEG | |
99 | Os | Delete os. Delete first require. Substitute invasive with intracranial. | *Need a reference. |
107 | Lateralizing | Lateralized, with switching sides, | Confusing what do you mean |
108 | poorer | less favorable | Still 40-50% Engle-I in bilateral independent temporal cases. |
109 | Add can highly predict | ||
110 | Delete with a high degree. | ||
112 | 5 | 3-5 | Many centers do 3 |
114 | In particular | Particularly | |
115 | Invasive | Intracranial | |
131 | MTS | Mesial temporal sclerosis (MTS) | Spell out the first use of abbreviation. |
147 | (interictal). Postictal) | Interictally. Postictally | |
Ictal, ictal, interictal | Ictally, ictally, interictally. | ||
156 | An | Delete | |
166 | And | Delete | |
192 | Invasive | Intracranial | |
227 | In lateralizing seizure onset | Delete | I am not sure of that information, needs reference. |
240 | Add coma after propofol | ||
246 | More | Delete | |
247 | Add which institution techniques is this | Differs among institution | |
253 | That play a significant role | Particularly | Avoid redundancy |
255 | Add lobe after temporal | ||
255 | Dominant (usually left) temporal lobe epilepsy Delete language dominant side. | ||
256 | Delete (most often, the left) | ||
258 | (typically the right) | Usually | Avoid redundancy. |
272 | Are | Is | |
275 | Invasive | Intracranial EEG recording (iEEG) | |
278 | Presurgical | Surgical | The fact it is a surgery by itself. |
279 | Invasive | Intracranial EEG | |
280 | Invasive | Intracranial | |
285 | Delete the | ||
287 | SEEG | Stereotactic EEG (SEEG) | |
288 | Polar | Orbitofrontal | |
298 | Add implantation at the end. | ||
312 | Delete In | ||
314 | They | The | |
404 | Delete including radiofrequency (RF) ablation and | As it has shown no efficacy. | |
411 | EEG | MRI | |
446 | Gamma Knife Radiosurgery | **Delete reference | |
452 | Resecting of | Dissecting | |
467 | Effective Delete as standard excisional therapy | ||
486 | ****In adult patient. VNS was approved by the FDA for pediatric patients this year. |
Thank you for this comprehensive review. I think your article will provide an excellent online educational tool for all of us including medical students and neurology residents.
Many thanks again.
Author Response
1/28/2017
Guest Editor
Warren W. Boling, MD
Department of Neurosurgery, Loma Linda University Medical Center, 25455 Barton Road, 108A, Loma Linda, CA 92354, USA
Ref.: Journal Brain Sciences (ISSN 2076-3425); Manuscript ID: brainsci-250676
Dear Dr. Boling,
We thank you and the reviewers for reviewing our manuscript. We have extensively revised the manuscript accordingly. Detailed below is our point-by-point response to the reviewers’ comments. We attach two copies of the new revised manuscript, a markup with all changes shown with Trach Changes On, and a clean copy. We hope that we have adequately addressed the concerns raised.
Please do not hesitate to contact me should you have any questions.
Sincerely,
Gholam Motamedi, MD
Georgetown University Hospital
Department of Neurology, PHC 7
3800 Reservoir Rd., NW
Washington, DC 20007
Tel: 202-444-4564
Fax: 877-245-1499
Email: Motamedi@georgetown.edu
Reviewers' comments and replies
Reviewer#1:
1) As the field of epilepsy surgery advances, minimally invasive theraputic and diagnostic procedures are being used more often. Therefore, I suggest softening the language used for epilepsy surgery by switching the word invasive recording to intracranial recording. In our center, the majority of our case are SEEG, which is minimally invasive.
- Done
2) I like the fact that you are bringing the new classification to your work. I suggest switching the parenthesis in line 63 to: refractory focal seizures with loss of awareness (complex partial seizures in previous classification).
- Done. We have also applied this elsewhere as needed.
3) Need a reference to this fact: Half temporal lobe epilepsy and all extra temporal lobe epilepsy will need invasive EEG recording. You can also say Non-lesional MRI (more specific).
- Done (new reference 11)
4) Suggest deleting reference 84:
- Done
5) In our experience MST is not as effective as excision surgery. Collective Data Supports Efficacy of Multiple Subpial Transection: Epilepsy Curr. 2002 Jul; 2(4): 108., doi: 10.1046/j.1535-7597.2002.t01-1-00039.x PMCID: PMC321030
- Done, and the paragraph further edited.
6) FDA Approves VNS Therapy Treatment for Drug-Resistant Epilepsy in Children as Young as Four Years Old Expanded access to VNS Therapy treatment offers new hope for children with drug-resistant epilepsy, London, June 29, 2017. Need to adjust your statement to include adult and pediatric population.
- Done
More suggested detailed editing (including medical and language editing) for your consideration.
- All suggestions applied
Reviewer#2:
Unfortunately, the paper consequently fails to properly flow, and instead becomes a body of loosely connected paragraphs, skipping briefly from topic to topic. There is no clear organization, no clear purpose, no focus, and thus no clear conclusions to be drawn. Disproportionate swaths of text are devoted to imaging modalities and surgical modalities with no connection, for example: Why would we bother to discuss corpus callosotomy if the earlier portions of the paper had been entirely devoted to focal epilepsy?
My initial suggestions based on the current manuscript would be to first determine a more narrow focus for the paper, for instance, the contribution of imaging to the surgical evaluation, or the current major surgical and nonsurgical intervention techniques. In such a case, the lack of details in the section on the new seizure classification and definitions of refractory epilepsy would be acceptable and those sections could easily be consolidated.
Alternately, one could make the focus slightly broader, focusing on the presurgical evaluation, but this requires a more methodical approach, including more detail on the clinical history, seizure semiology and classification, an expansion on the principles of the "cortical zones", and then could tie in the utility of imaging modalities and supporting tests (Wada and neuropsychological evaluation).
- This point is well taken. We agree with the comment. This happened since initially the scope of this article –as part of a special issue- was unclear. After discussing the options with the Editor, and considering the scope of other articles included in this special issue, we chose to keep this article more comprehensive as an overall approach to surgical management of epilepsy. Therefore, we have extensively revised and restructured the manuscript accordingly.
- The revision includes removing and adding new paragraphs; changing and adding new titles and subtitles, adding a whole new section at the end on outcome measures, adding 17 new references, and importantly keeping the flow of manuscript flow of the article; as part of this, the title was also modified to reflect the content better. Aa a general topic one may not be able to cover all relevant issues in details but we believe the new revised version has addressed the concerns as much as possible and reads better.
Reviewer#3:
….. this article does not provide a guide. The article describes techniques without providing a route map. Therefore I would recommend that the authors consider a different style of paper ….. Could the authors, therefore, describe a route map for the management of this group?
- The revised version has addressed these concerns as much as possible and reads better, as described above. Please see the reply to reviewer#2.
If there is a need for WADA and invasive electrode monitoring what is the risk to an individual or are multiple invasive procedures not required. The authors frequently quote mortality for the various diagnostic procedures as acceptable. But at 1%, that appears to me to be quite high.
- Done. We have thoroughly edited the sections on Wada and neuropshological testing with special attention to safety and efficacy concerns. Further, a whole new section on postsurgical outcome has been added.
There is also an absence of discussion about neuropsychological deficits that may follow the diagnostic procedures or the surgery.
- Done. Please see the above reply.
Reviewer 2 Report
This review article attempts to cover a vast number of topics, all of which are relevant to the diagnosis and surgical treatment of epilepsy. Unfortunately, the paper consequently fails to properly flow, and instead becomes a body of loosely connected paragraphs, skipping briefly from topic to topic. There is no clear organization, no clear purpose, no focus, and thus no clear conclusions to be drawn. Disproportionate swaths of text are devoted to imaging modalities and surgical modalities with no connection, for example: Why would we bother to discuss corpus callosotomy if the earlier portions of the paper had been entirely devoted to focal epilepsy?
My initial suggestions based on the current manuscript would be to first determine a more narrow focus for the paper, for instance, the contribution of imaging to the surgical evaluation, or the current major surgical and nonsurgical intervention techniques. In such a case, the lack of details in the section on the new seizure classification and definitions of refractory epilepsy would be acceptable and those sections could easily be consolidated.
Alternately, one could make the focus slightly broader, focusing on the presurgical evaluation, but this requires a more methodical approach, including more detail on the clinical history, seizure semiology and classification, an expansion on the principles of the "cortical zones", and then could tie in the utility of imaging modalities and supporting tests (Wada and neuropsychological evaluation).
Author Response

(The authors gave the same response as above.)

Reviewer 3 Report
Diagnosis and Surgical Treatment of Epilepsy
This is not the first review of the surgical treatment of epilepsy I have reviewed this year. I am assuming that the authors are experienced in the assessment and consideration of patients for neurosurgery for epilepsy. This article provides a comprehensive description of the assessment of patients with epilepsy. There is some description of the surgical procedures that are available but this is less comprehensive than the diagnostic discussion. I will assume this is because the speciality of the authors is not surgical.
Whilst it is helpful to have another overview of the diagnosis and treatment of patients who are refractory to epilepsy, my opinion is that this article does not provide a guide. The article describes techniques without providing a route map.
Therefore I would recommend that the authors consider a different style of paper. If surgery is not their expertise, they should review the need for those elements of the paper.
The challenge for the physician is the management of patients who are refractory to treatment. Could the authors, therefore, describe a route map for the management of this group? Perhaps after failing second and third line therapies what investigations should be carried out? What are the morbidity and mortality of each investigation? How does this inform further investigations? I would expect that the investigations would become increasingly invasive with higher complications. If there is a need for WADA and invasive electrode monitoring what is the risk to an individual or are multiple invasive procedures not required.
The authors frequently quote mortality for the various diagnostic procedures as acceptable. But at 1%, that appears to me to be quite high. There is also an absence of discussion about neuropsychological deficits that may follow the diagnostic procedures or the surgery. There is also no counter discussion about living with intractable epilepsy. Perhaps the comparative data is not available, but the authors could discuss clinical challenges moving forward.
In summary, I would like the authors to consider what a route map would look like if they were presented today with a patient with intractable epilepsy. How would the discussion with the patient appear, if they were to recommend a non-pharmacological approach? What are the risks and benefits?
Also, where are the gaps in the data regarding a non-pharmacological approach?
If the authors were able to provide this, I think this would separate this paper from the current review papers on this subject.
Author Response

(The authors gave the same response as above.)

Round 2
Reviewer 3 Report
This manuscript is much improved and now has a direction and a balance that was missing from the earlier script. I would now have no criticisms of this current document
Author Response
3/5/2017
Guest Editor
Warren W. Boling, MD
Department of Neurosurgery, Loma Linda University Medical Center, 25455 Barton Road, 108A, Loma Linda, CA 92354, USA
Ref.: Journal Brain Sciences (ISSN 2076-3425); Manuscript ID: brainsci-250676
Dear Dr. Boling,
Thank you and the reviewers for reviewing our manuscript. We have revised the manuscript accordingly. Detailed below is response to a very few of the comments.
Please do not hesitate to contact me should you have any questions.
Sincerely,
Gholam Motamedi, MD
Georgetown University Hospital
Department of Neurology, PHC 7
3800 Reservoir Rd., NW
Washington, DC 20007
Tel: 202-444-4564
Fax: 877-245-1499
Email: Motamedi@georgetown.edu
Reviewers' comments and replies
Reviewers # 3
This manuscript is much improved and now has a direction and a balance that was missing from the earlier script. I would now have no criticisms of this current document
- Thank you.
Editor’s comments
All comments and editing have been applied except for the following:
- Line 276: The statement “Wada test is only indicated in patients who may undergo temporal lobectomy.” was modified by removing “only”. Wada test does not add to the presurgical planning in extra temporal lobe resections.
- Line 302: “Therefore, some patients with TLE, including those with left-sided TLE, …”. Here the “left-sided” is the intended statement and was kept.
- Line 282: the statement “but more critically in patients at risk of postsurgical memory loss” was kept as it is emphasizing the more critical role of the Wada test i.e., with the increasing use of fMRI and its concordance with Wada language test, there is decreasing need for Wada as a tool to determine the language dominance. Therefore, Wada test plays a more critical role in terms of memory concerns.
- Line 639: 10% is correct. Please see the abstract below:
N Engl J Med. 2001 Aug 2;345(5):311-8.
A randomized, controlled trial of surgery for temporal-lobe epilepsy.
Wiebe S, Blume WT, Girvin JP, Eliasziw M; Effectiveness and Efficiency of Surgery for Temporal Lobe Epilepsy Study Group.
RESULTS: At one year, the cumulative proportion of patients who were free of seizures impairing awareness was 58 percent in the surgical group and 8 percent in the medical group (P<0.001). The patients in the surgical group had fewer seizures impairing awareness and a significantly better quality of life (P<0.001 for both comparisons) than the patients in the medical group. Four patients (10 percent) had adverse effects of surgery. One patient in the medical group died.